

# Seismic amplitude response to internal heterogeneity of mass-transport deposits

Jonathan Ford[1], Angelo Camerlenghi[1], Francesca Zolezzi[2], and Marilena Calarco[2]

[1]National Institute of Oceanography and Applied Geophysics – OGS, Trieste, Italy
[2]RINA Consulting, Genova, Italy

**Correspondence:** J. Ford (jford@inogs.it)

**Abstract.** Compared to unfailed sediments, mass-transport deposits are often characterised by a low-amplitude response in single-channel seismic reflection images. This 'acoustic transparency' amplitude signature is widely used to delineate mass-transport deposits and is conventionally interpreted as a lack of coherent internal reflectivity due to a loss of preserved internal structure caused by mass-transport processes. In this study we examine the variation in the single-channel seismic response

with changing heterogeneity using synthetic 2-D elastic seismic modelling. We model the internal structure of mass-transport deposits as a two-component random medium, using the lateral correlation length ($a_x$) as a proxy for the degree of internal deformation, whilst maintaining approximately constant internal reflectivity with increasing deformation. For a controlled single-source synthetic model a reduction in observed amplitude with reduced $a_x$ is consistently observed across a range of vertical correlation lengths ($a_z$). For typical AUV sub-bottom profiler acquisition parameters, in a simulated mass-transport

deposit with realistic elastic and geostatistical properties, we find that when $a_x \approx 1$ m, recorded seismic amplitudes are, on average, reduced by $\sim 15\%$ relative to unfailed sediments ($a_x \gg 10^3$ m). We also observe that deformation significantly larger than core-scale ($a_x > 0.1$ m) can generate a significant amplitude decrease. These synthetic modelling results should discourage interpretation of the internal structure of mass-transport deposits based on seismic amplitudes alone, as 'acoustically transparent' mass-transport deposits may still preserve coherent, metre-scale internal structure. In addition, the minimum scale

of heterogeneity required to produce a significant reduction in seismic amplitudes is likely much larger than the diameter of sediment cores, meaning that 'acoustically transparent' mass-transport deposits may still appear well-stratified and undeformed at core-scale.

## 1 Introduction

The internal structure of mass-transport deposits (MTDs) preserves information on the flow type, post-failure dynamics and

emplacement of subaqueous mass-movements (Mulder and Cochonat, 1996; Lucente and Pini, 2003; Ogata et al., 2016; Sobiesiak et al., 2016). Geophysical imaging of internal structure can therefore play an important role in constraining the geohazard potential from mass-movements such as submarine landslides and debris flows (e.g., Strasser et al., 2011; Pini et al., 2012; Ogata et al., 2019; Karstens et al., 2019). MTDs are often identified, delineated and classified based on their distinctive seismic character (or 'echofacies') in seismic reflection data (Moscardelli and Wood, 2008; Alves et al., 2014; Clare et al., 2019).





Due to their non-conformal upper and lower surfaces, MTDs are frequently bounded by high-amplitude, laterally continuous top and basal reflectors (Frey-Martinez et al., 2005). Their internal structure, instead, is often reported to have a characteristic low-amplitude seismic response compared to unfailed sediments. This seismic character has previously been described as 'semi-transparent' (Piper et al., 1997; Moernaut et al., 2020), 'acoustically transparent' (Talling et al., 2010; Hunt et al., 2021), 'low reflectivity' (Sawyer et al., 2009), 'transparent-to-chaotic' (Posamentier and Martinsen, 2011) or similar. To date,
however, the precise geophysical mechanisms that control this low-amplitude internal seismic response have received little attention from the marine geohazard community. Previous studies have invoked mechanisms such as the more uniform physical properties of sediments within MTDs due to, e.g., over-compaction or fine-scale mixing during sediment transport (Posamentier and Kolla, 2003; Shipp et al., 2004; Sawyer et al., 2009). Others have implied that the loss of coherent seismic character results from internal disaggregation, indicating a debris flow-like deposit lacking coherent internal structure (e.g., Diviacco
et al., 2006; Hunt et al., 2021). In other words, many studies make an implicit assumption that the internal seismic character of MTDs can be directly related to the preservation of internal structure (or lack thereof).

One indication that the seismic amplitude response is not straightforwardly related to preserved internal structure is that modern high-resolution geophysical datasets, particularly 3-D seismic volumes, have revealed previously unresolvable organisation and internal structure within MTDs as small as metre-to-decametre scale (Bull et al., 2009; Gafeira et al., 2010; Alves
and Lourenço, 2010; Bellwald and Planke, 2018; Badhani et al., 2020; Barrett et al., 2021). Another indication is that sediment core samples retrieved from seismically 'transparent' bodies can sometimes show little evidence of internal deformation at core-scale (10s of cm and lower) (Expedition 316 Scientists, 2009; Strasser et al., 2011; Sammartini et al., 2021, Fig. 2, this study). This apparent contradiction prevents both qualitative and quantitative correlation between core and seismic data inside MTDs. Finally, field evidence from high-resolution outcrop studies often shows a high degree of internal structural organisation
preserved within exhumed 'fossil' MTDs, over sub-centimetre to kilometre scale lengths (Lucente and Pini, 2003; Ogata et al., 2016).

## 1.1 Aims and objectives

To infer geological structure from the seismic response, and to properly correlate seismic and core observations, we need an improved understanding of the geophysical controls on the seismic amplitude response in MTD-like (i.e., complex and strongly
heterogeneous) geology. To address this, we model the single-channel seismic response of a range of geological models that aim to approximate—in a general sense—the heterogeneous internal structure within MTDs. We model the internal structure of MTDs as anisotropic, two-component (binarised) random fields, and vary the lateral correlation length to simulate changing degrees of structural deformation. We perform 2-D elastic finite difference modelling of the seismic response at realistic sub-bottom profiler bandwidths, with two experimental setups: i) a controlled single-source synthetic experiment based on an
simple homogeneous-heterogeneous four layer marine model, and ii) a realistic AUV multi-source synthetic profile based on an alongslope sub-bottom profile from a 2017 Black Sea geohazard survey. By isolating the effect of internal deformation from lithological and petrophysical alteration by mass-movement processes, we aim to better understand the possible contribution of changing heterogeneity to the amplitude of the single-channel seismic response.





## 2 Black Sea geohazard survey case study

A marine geohazard survey was carried out in the Black Sea in 2017 to support construction of a planned pipeline. The survey comprised geophysical data (multi-channel seismic, bathymetry and single-channel 'Chirp' sub-bottom seismic profiles), gravity and piston core sampling and geotechnical investigation with cone-penetration tests (CPT). The main study area was a submarine canyon incised into the continental slope, in water depths between approximately $-100$ to $-1800$ m. The results were used to identify and characterise the main geohazards in the study area such as faults, sediment failure and fluid

migration, and to inform parameters used in engineering studies for pipeline design. Data were acquired from two platforms: a traditional survey vessel and an accompanying autonomous underwater vehicle (AUV). The AUV simultaneously acquired high-resolution bathymetry (approx. $2 \times 2$ m resolution in the area considered in this study) and a dense grid of 2-D sub-bottom profiles (approx. $100 \times 700$ m longitudinal/cross-line spacing in the area considered in this study).

From the bathymetry and AUV Chirp data we mapped a large MTD emplaced in the centre of the canyon (Fig. 1). The MTD

lies between approx. $-400$ and $-1100$ m water depth, buried under a thin ($< 2$ m thick) sediment drape. It has a maximum width of approx. $1.8$ km, a maximum thickness of approx. $35$ m and runs out for a mapped length of $> 23$ km down the canyon. The total runout length and volume are unknown because the survey area does not cover the entire MTD body. Several sediment cores and CPT measurements were made inside and in the immediate vicinity of the MTD (see Fig. 1 and Figs. S1 and S2).

AUV Chirp profiles intersecting the MTD show a generally consistent seismic character (Figs. 1b and 1c). The apparent basal surface consistently appears as a coherent, high-amplitude seismic horizon. The apparent top surface is less well defined, but is generally visible and topped by a sediment drape characterised by high-amplitude, sub-parallel reflectors. The internal character of the MTD is generally low-amplitude relative to the surrounding unfailed, well-stratified sediments—a classic apparently 'acoustically transparent' seismic character. The root-mean-squared (RMS) average envelope amplitude within the

body is approximately half of the RMS envelope amplitude of the unfailed background sediments (Fig. 1d-e). A sediment core taken inside the MTD shows clear stratification, with little evidence of disturbed bedding or deformation structures within the interval that corresponds to the 'acoustically transparent' MTD (Fig. 2).

## 3 Methodology

### 3.1 Single-source synthetic experiment

The single-source 2-D model has four layers: a water layer and three sediment layers comprising a heterogeneous layer bounded by two homogeneous layers (Fig. 3a). The heterogeneous layer is an anisotropic, two-component (binarised) 2-D random medium with exponential autocorrelation, defined by its lateral and vertical correlation lengths, $a_x$ and $a_z$. The elastic parameters of the component sediment lithologies and water layer are listed in Table 1. The model is discretised on a regular grid with dimensions $801 \times 801$ grid points and sampling interval $0.025 \times 0.025$ m, for a total extent of $40 \times 40$ m.





**Figure 1.** Example from the 2017 Black Sea geohazard survey of an MTD where the internal seismic amplitude response is appreciably lower compared to the unfailed sediments ('acoustic transparency'). a) Map showing location of the SBPs (b and c), sediment core and CPT measurement relative to the interpreted extent of the MTD. Background bathymetry from GEBCO Compilation Group (2021). b) Downslope-oriented and c) alongslope-oriented AUV SBPs intersecting the deposit. d) and e) Amplitude analysis histograms. 'MTD' corresponds to mass-transport deposit, 'SBP' corresponds to sub-bottom profile.




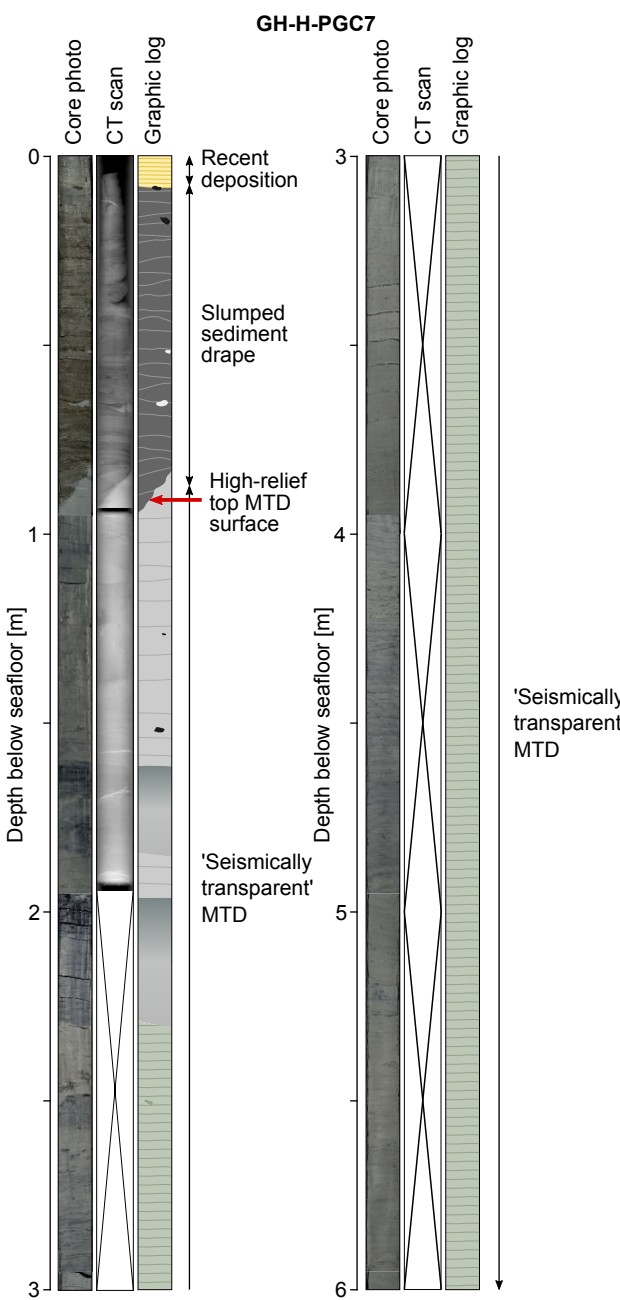

**Figure 2.** Piston gravity core GH-H-PGC7 (see Fig. 1 for location). Core photograph, X-ray computed tomography image (note: only first 2 m are scanned) and interpreted graphical log. The 'seismically transparent' MTD corresponds to the mass-transport deposit labelled in Fig. 1b.





**Table 1.** Elastic parameters for the water layer and the two sediment lithologies in the single-source synthetic experiment.

|  | P-wave velocity | S-wave velocity | Density |
| --- | --- | --- | --- |
| Water | $1500 \text{ ms}^{-1}$ | — | $1000 \text{ kgm}^{-3}$ |
| Lithology 1 | $1600 \text{ ms}^{-1}$ | $800 \text{ ms}^{-1}$ | $1200 \text{ kgm}^{-3}$ |
| Lithology 2 | $1700 \text{ ms}^{-1}$ | $850 \text{ ms}^{-1}$ | $1400 \text{ kgm}^{-3}$ |

The seismic source is located within the water layer (Fig. 3a). The source is an impulsive pressure source with Ricker wavelet with dominant frequency 3.5 kHz, to match common 'Chirp' sub-bottom profiler source bandwidths used for high-resolution seismic profiling of shallow sediments (Gutowski et al., 2002). The receiver records the pressure wavefield and is located coincident with the source, laterally offset by one grid point to avoid numerical artefacts associated with a co-located source and receiver in elastic modelling.

The finite-difference modelling scheme is elastic, 4th order in space, 2nd order in time. The modelling uses absorbing boundaries (sponge layers) with width 100 grid points on all four edges of the grid. For the elastic parameters in this experiment (Table 1) the critical timestep is $\Delta t_c = 0.0075$ ms (Carcione, 2014). The timestep used is $\Delta t = 0.9\Delta t_c = 0.0067$ ms, to ensure stability. The modelling is run for 21.8 ms (2912 timesteps), long enough to record a P-P reflection with the average P-wave velocity from the deepest part of the model directly beneath the source point.

**3.2    Realistic multi-source synthetic experiment**

The realistic multi-source 2-D synthetic model is based on the alongslope example profile from the 2017 Black Sea geohazard survey (Fig. 1c). The model consists of a homogenous water layer with a variable-depth waterbottom, below which the background (unfailed) sediment layer is modelled as a two-component (binarised) 1-D random medium with exponential autocorrelation, hung from the waterbottom. This simulates conformal, parallel bedding, similar in character to the background

unfailed sediments generally observed in sub-bottom profiles from the study area (e.g., Fig. 1b and Fig. 1c).

     Realistic elastic parameters for the two component sediment lithologies are derived from multi-sensor core logging (MSCL) measurements from four sediment cores in the study area (GH-H-PGC7, GH-H-PGC8, GH-H-JPC4A and GH-H-JPC5A). P-wave velocity and density logs and crossplots are documented in Fig. S1. We define the P-wave velocity and density of each component lithology as approximately one standard deviation in each direction from the mean values for the two parameters.

We assume that the ratio of P- to S-wave velocity in the sediments is 2. Elastic parameters for the component sediment lithologies are listed in Table 2. It should be noted that we do not aim to estimate realistic elastic parameters of distinct lithologies from the study area, rather we use the distribution of the values to give plausible reflectivities (i.e., the *contrast* in elastic parameters) within the sediment column.

     Realistic vertical geostatistical parameters for the sediments in the model are derived from cone penetration tests (CPT) from

the study area. The cone-tip resistance log for CPT location GH-T-PCPT7, along with the spatial autocorrelation function, is documented in Fig. S2. We define the vertical correlation length in the sediments as $a_z = 0.05$ m, approximately consistent



with the measured spatial autocorrelation functions (Fig. S2c). It should be noted that we do not aim to replicate exactly the realistic vertical geostatistical parameters of the sedimentary column in the study area, rather we use the autocorrelation function to estimate plausible 'bed thicknesses' (closely related to $a_z$) for this simplified two lithology model.

The MTD zone is modelled as an anisotropic, two-component (binarised) 2-D random medium with exponential autocorrelation, located close to the seafloor but partially covered by a thin drape of background sediment (Fig. 5a). Elastic parameters for the component lithologies are identical to the surrounding unfailed sediments (Table 2). The MTD zone random medium uses the same random seed as the unfailed sediments and is offset vertically from the waterbottom so that in the pre-failure state (arbitrarily set as $a_x = 10^7$ m) the beds are parallel and continuous with the surrounding unfailed sediments (Fig. 5b).

We simulate increasing post-failure deformation by progressively decreasing the lateral correlation length until the random medium is isotropic ($a_x = a_z = 0.05$ m). An example of a post-failure MTD model with lateral correlation length $a_x = 50$ m is shown in Fig. 5c.

    The seismic source locations follow a 'flight path' similar to a realistic AUV acquisition. For the data acquired in the 2017 Black Sea geohazard survey (Fig. 1) the AUV flight path was targetted around 40 m above the seafloor. We replicate a similar

profile by placing the shot locations along a smoothed waterbottom, shifted up by 40 m, with horizontal shot spacing 2 m (Fig. 5a). The source is an impulsive pressure source with Ricker wavelet with dominant frequency 1.1 kHz. The receiver records the pressure wavefield and is located co-incident with the source, laterally offset by one grid point to avoid numerical artefacts associated with a co-located source and receiver in elastic modelling.

    The finite-difference modelling scheme is elastic, 4th order in space, 2nd order in time. The global model is discretised

on a regular grid with sampling interval $0.1 \times 0.1$ m, for a total extent of $5000 \times 160$ m. For computational efficiency, the global model is partitioned into sub-models for each shot. Each sub-model is centred on the shot location with lateral padding zones of 50 m width on either side of the source location (total width 100 m for shots in the centre of the model). Because the sub-model is cut above the source, the number of vertical grid points depends on the source depth. A typical sub-model from a shot in the centre of the global model has grid size $1001 \times 1001$ grid points. The modelling uses absorbing boundaries

(sponge layers) with width 100 grid points on all four edges of the grid. For the elastic parameters in this experiment (Table 1) the critical timestep is $\Delta t_c = 0.033$ ms (Carcione, 2014). The timestep used is $\Delta t = 0.9 \Delta t_c = 0.030$ ms, to ensure stability. The modelling is run for long enough to record a P-P reflection, with the average velocity from the deepest part of the model directly beneath the source point. A typical sub-model from a shot in the centre of the global model is run for approx. 120 ms (3625 timesteps).

## 4   Results

### 4.1   Single-source synthetic experiment

We generate realisations of the single-source synthetic experiment with lateral correlation lengths $10^{-3} \leq a_x \leq 10^4$ m ($n = 8$) and vertical correlation lengths $10^{-2} \leq a_z \leq 1$ m ($n = 5$) within the heterogeneous layer. For each distinct combination of parameters, $[a_x, a_z]$, we generate an ensemble of realisations ($n = 10$) by varying the random seed used to generate the random




**Table 2.** Elastic parameters for the water layer and the two component sediment lithologies in the realistic multi-source synthetic experiment. Sediment parameters are based on multi-sensor core logging data from sediment cores located in the study area (see Fig. S1).

|  | P-wave velocity | S-wave velocity | Density |
|---|---|---|---|
| Water | $1500 \text{ ms}^{-1}$ | — | $1000 \text{ kgm}^{-3}$ |
| Lithology 1 | $1300 \text{ ms}^{-1}$ | $650 \text{ ms}^{-1}$ | $1900 \text{ kgm}^{-3}$ |
| Lithology 2 | $1400 \text{ ms}^{-1}$ | $700 \text{ ms}^{-1}$ | $2100 \text{ kgm}^{-3}$ |

field, giving a total of $n = 400$ distinct models and modelling runs. Fig. 3 shows the single-source synthetic experiment model geometry and one realisation of a single-source synthetic model with correlation lengths $[a_x, a_z] = [1, 0.05]$ m.

We run the forward modelling for all realisations in parallel on a desktop PC with a total of 40 vCPU cores. Fig. 4a shows the envelope of the modelled traces, recorded in two-way traveltime (TWTT) at the receiver, for a selection of the models with vertical correlation length $a_z = 0.05$ m. Reducing the lateral correlation length from $a_x = 1000$ m to $a_x = 0.1$ m sys-

tematically reduces the amplitudes recorded within the heterogeneous layer. A decaying coda is seen beneath the base of the heterogeneous layer, presumably associated with multiple reflections and scattering within the heterogeneous layer. For the longest lateral correlation lengths, the amplitudes in this coda are very low compared to within the heterogeneous layer. As the lateral correlation lengths decrease, the amplitudes in the coda systematically increase, until they are comparable to the amplitudes observed within the heterogeneous layer.

Fig. 4b shows the root-mean-squared (RMS) envelope amplitude of the modelled traces within the TWTT window corresponding to the heterogeneous layer, against lateral correlation length, $a_x$. For all vertical correlation lengths tested, a reduction in lateral correlation length causes a reduction in RMS amplitude. The most rapid drop in recorded amplitude occurs when lateral correlation lengths are $1 \le a_x \le 10^{-2}$ m. The RMS vertical-incidence acoustic reflectivity remains approximately constant across all model realisations.

**4.2 Realistic multi-source synthetic experiment**

We generate realisations of the realistic multi-source synthetic experiment with lateral correlation lengths $5 \times 10^{-2} \le a_x \le 10^7$ m ($n = 8$) within the MTD zone. For each lateral correlation length we generate an ensemble of realisations ($n = 5$) by varying the random seed used to generate the random fields, giving a total of $n = 40$ distinct global models and modelling runs. The model geometry and two realisations of the global model, one representing the undeformed, pre-failure state ($a_x = 10^7$ m, i.e.,

parallel bedding) and one representing a deformed, post-failure MTD ($a_x = 50$ m), are shown in Fig. 5.

We run the forward modelling for all realisations in parallel on a desktop PC with a total of 40 vCPU cores. Fig. 6 shows examples of simulated sub-bottom profiles for sources $1000 \le a_x \le 4000$ m, recorded in TWTT at the receiver, for three models: one representing the undeformed, pre-failure state ($a_x = 10^7$ m, i.e., parallel bedding; Fig. 6a), one representing a partially deformed MTD ($a_x = 10$ m; Fig. 6b) and one representing a strongly disrupted MTD ($a_x = 10^{-1}$ m; Fig. 6c). We also

plot the envelope amplitude of single traces from inside and outside the MTD zone for all realisations of each set of parameters,



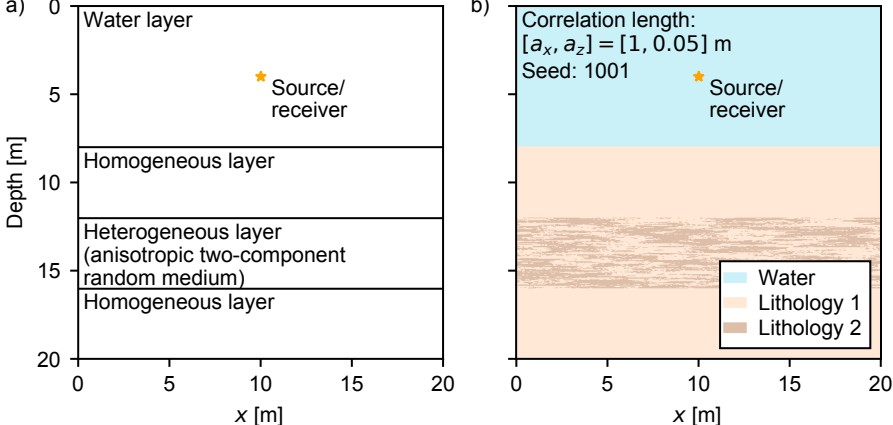

**Figure 3.** Single-source synthetic experiment. a) Model geometry. Both homogeneous sediment layers are composed entirely of Lithology 1. The heterogeneous layer is an anisotropic, two-component exponential random medium composed of equal parts Lithology 1 and Lithology 2. Elastic parameters are listed in Table 1. The coincident seismic source and receiver location within the water layer is marked (yellow star). b) A single realisation of the model showing the spatial distribution of Lithology 1 and Lithology 2 within the heterogeneous layer (correlation lengths $[a_x, a_z] = [1, 0.05]$ m, seed 1001).

along with the ensemble RMS average traces. Visually, the modelled sub-bottom profiles show increasing 'transparency' (i.e., an apparent decrease in average amplitude and the lateral continuity of reflectors) with decreasing lateral scale length. As in the single-source synthetic experiments, an apparent coda is visible as a noisy zone beneath the MTD for the shortest lateral scale lengths.

Fig. 6d shows the RMS amplitude of the modelled traces within and below the MTD zone. For all vertical correlation lengths tested, a reduction in lateral correlation length causes a reduction in RMS amplitude. The average vertical-incidence acoustic impedance contrast remains approximately constant across all model realisations.

## 5   Discussion

Our modelling of single-channel seismic experiments in two-component, anisotropic random media models shows a significant
average amplitude reduction with decreasing lateral correlation length, despite the average reflectivity within the random media remaining approximately constant (Figs. 4 and 6). We observe this effect with two synthetic single-channel seismic reflection experiments:

1. A simplified single-source model with a heterogeneous random media layer (Section 3.1).

2. A more realistic multi-source model, with an AUV-style sub-bottom profiler acquisition and geologically plausible elastic
and geostatistical parameters, with a random media zone representing an MTD, based on a sub-bottom profile from the 2017 Black Sea geohazard survey (Section 3.2).





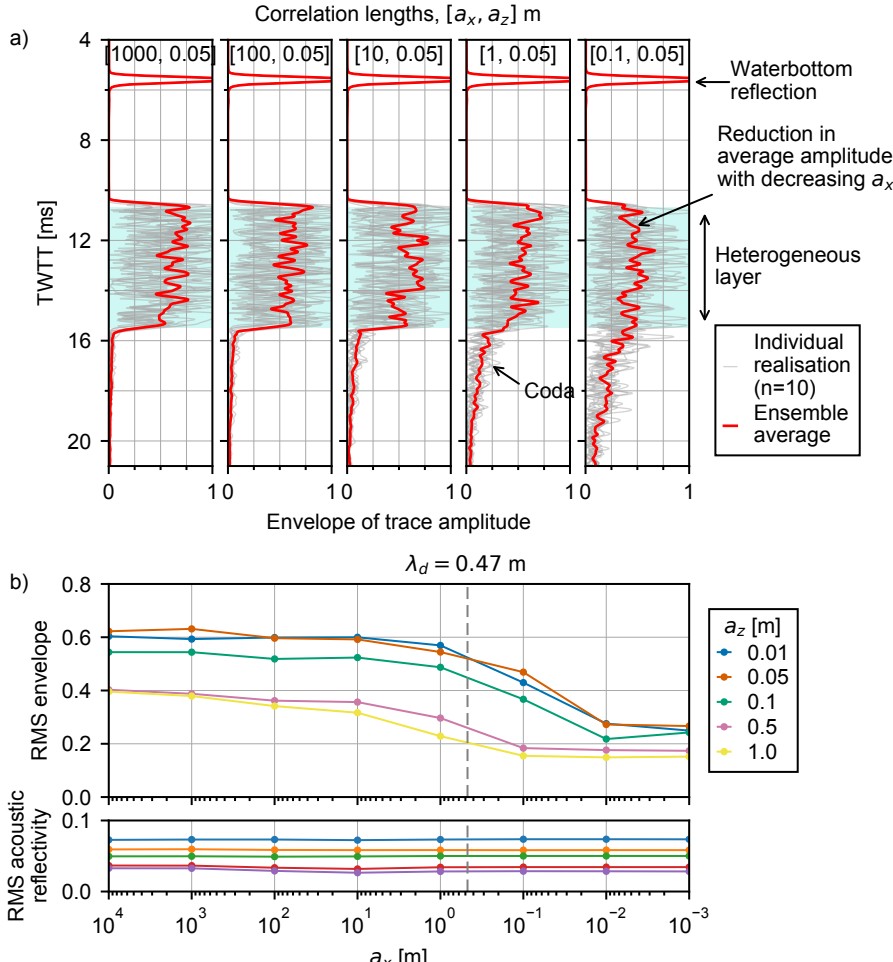

**Figure 4.** Single-source synthetic experiment results. a) Envelope of trace amplitude for $n = 10$ multiple realisations (grey) and the RMS envelope of all realisations (red) for fixed vertical correlation length $a_z = 0.05$ m and lateral correlation lengths $a_x = \{1000, 100, 10, 1, 0.1\}$ m (from left to right). The two-way traveltime (TWTT) extent of the heterogeneous layer is shaded in blue. b) (Top) RMS envelope within the heterogeneous zone against lateral correlation length, $a_x$, grouped by vertical correlation length, $a_z$. (Bottom) RMS vertical incidence acoustic reflectivity within the heterogeneous zone. $\lambda_d$ shows the dominant wavelength of the 3.5 kHz seismic source in the water layer.



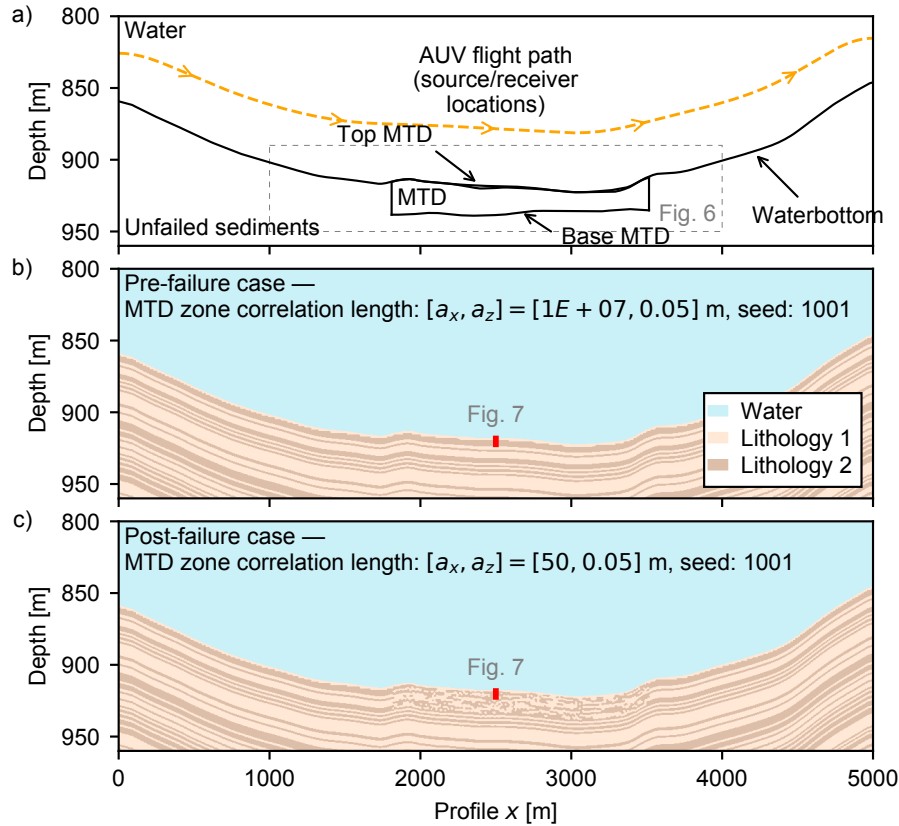

**Figure 5.** Realistic multi-source synthetic model based on the alongslope AUV sub-bottom profile from the 2017 Black Sea geohazard survey. a) Model geometry. The water layer is homogeneous. The unfailed background sediments and the MTD zone are two-component exponential random media comprising equal parts Lithology 1 and Lithology 2 (see Section 3.2 for details of the random fields). Elastic parameters of the component lithologies are listed in Table 2. The source locations along the AUV flight path, approx. 40 m above the waterbottom, are marked by the yellow dashed line (source interval 2 m). Two realisations of the model are shown, both with seed 1001. b) MTD zone in pre-failure state (lateral correlation length is equivalent to unfailed sediments, $a_x = 10^7$ m); c) MTD zone in post-failure state (shorter lateral correlation length, $a_x = 50$ m). The location and extent of the synthetic sediment core in Fig. 7 is shown in red. 'MTD' corresponds to mass-transport deposit.



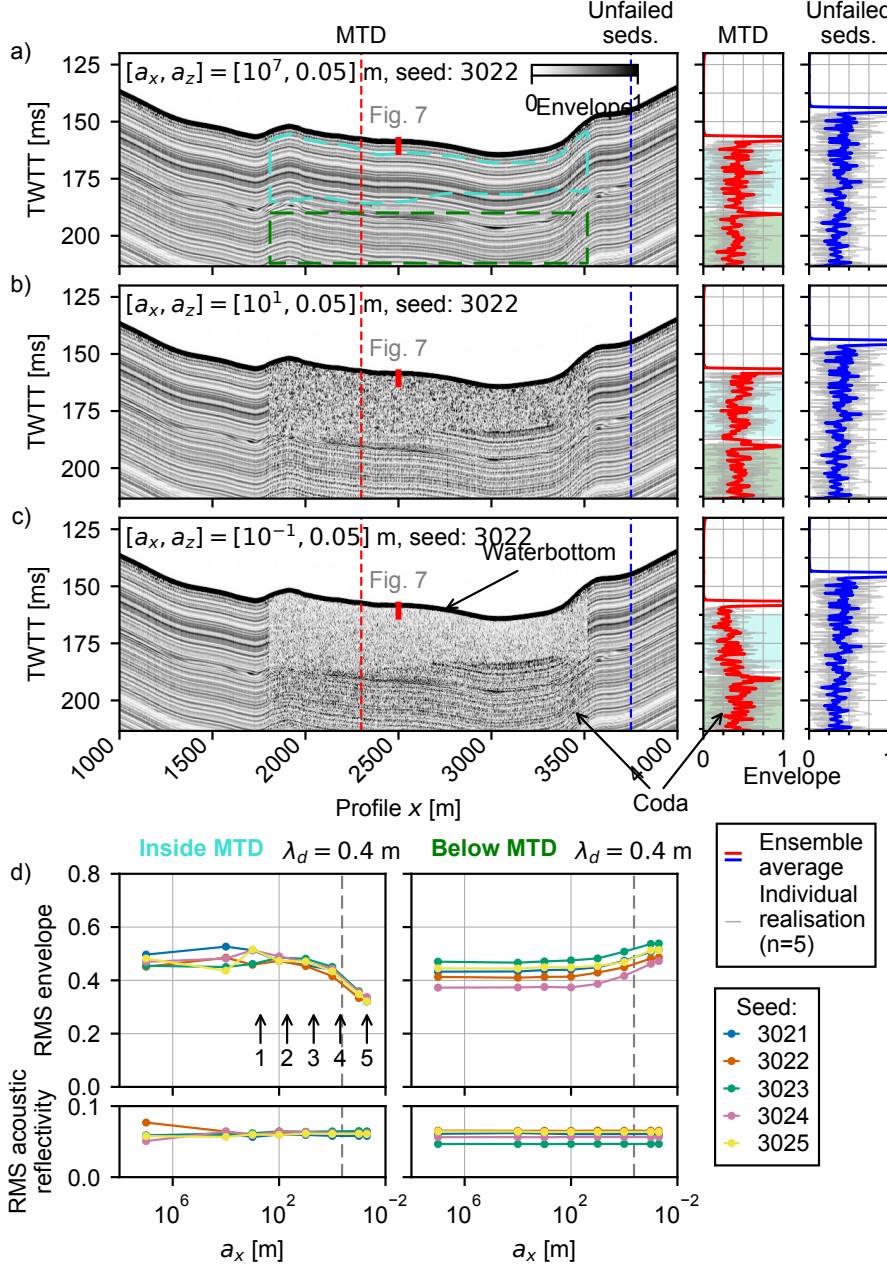

**Figure 6.** Realistic multi-source synthetic experiment results. a-c) Synthetic AUV sub-bottom profiles of the seismic response (envelope of trace amplitude) to progressively decreasing lateral correlation length, $a_x$ (i.e., increasing deformation). Right hand side plots show trace envelopes for the individual model realisations (grey) and the RMS amplitude of all realisations for traces intersecting the MTD (red) and unfailed sediments (blue). The two amplitude analysis windows are highlighted in cyan (internal MTD window) and green (below MTD deposit window). d) (Top) RMS of the trace envelope and (bottom) RMS vertical incidence acoustic reflectivity against lateral correlation length ($a_x$) for all realisations, both inside the MTD (left; cyan window in a) and below the MTD (right; green window in a). $\lambda_d$ shows the dominant wavelength of the 1.1 kHz seismic source in the water layer. 'MTD' corresponds to mass-transport deposit.





The results of these modelling experiments show that geological heterogeneity, in particular lateral heterogeneity, has a strong control on the average recorded seismic amplitudes in single-channel seismic data. The observed amplitude reduction effect is somewhat unexpected, as with a simple model for the seismic response, the average amplitude response should be approxi-

mately proportional to the average reflectivity. Interpretation of single- and multi-channel seismic images very often assumes a simple 'convolutional' model whereby the seismic image represents the zero-offset acoustic reflectivity convolved with the source wavelet. This would imply an average amplitude response that is approximately proportional to the average reflectivity. The systematic decrease in average amplitude with increasing heterogeneity means that in strongly heterogeneous media (i.e., complex, strongly deformed geobodies), this acoustic, 'convolutional' model is no longer a good approximation of the

subsurface reflectivity (i.e., the internal structure of the geobody).

## 5.1 Seismic amplitude response of heterogeneous media

The two synthetic modelling experiments demonstrate a systematic amplitude decrease with decreasing lateral correlation length (Figs. 4 and 6), but the results do not conclusively identify a geophysical mechanism that might cause this effect.

We do expect that seismic tuning, where the distance between two distinct reflectors is around the dominant source wave-

length, will have an effect on the recorded amplitudes (Chung and Lawton, 1999). Constructive interference will increase—and destructive interference decrease—the recorded amplitudes. In our experiments, the primary control on the vertical distance between reflectors is the vertical correlation length, roughly equivalent to the average 'bed thickness' (the lateral correlation length has a minor control, where bed thickness tends to zero at the lateral terminations). These variations in bed thickness may explain a small component of the variation in seismic amplitude with changing lateral correlation length. The single-source

synthetic experiment, however, reproduces significant, sustained amplitude decrease with decreasing lateral correlation length over a two-orders-of-magnitude variation in vertical correlation length (Fig. 4b). The fact that we observe the amplitude reduction effect over this range implies that seismic tuning between thin beds is not the dominant control on the average amplitude response. This also means that the heterogeneity-induced amplitude reduction effect should not be strongly dependent on the dominant wavelength of the seismic source.

Real-world seismic reflection experiments commonly show amplitude variation with changing incidence angle between the seismic wavefront and the reflector, caused by partitioning between transmitted and reflected P- and S-wave energy across an interface in elastic media (Shuey, 1985). This is often observed in multi-channel seismic data as an 'amplitude variation with offset' (AVO) effect (Avseth et al., 2010). In zero-offset data (i.e., single-channel data), however, the incidence angle of primary reflections should always be normal to the interface. Altering the geostatistical properties of the random media

does however change the *distribution* of the dips of the interfaces within the heterogeneous zone. In the far-field, the seismic wavefront can be approximated as a plane-wave. In the case of parallel horizontal bedding, the reflectors will be tangential to the wavefront, and thus generate an 'ideal' reflection. In the opposite extreme, where bedding is parallel and vertical, a surface seismic reflection experiment would not image these interfaces, even if there is high reflectivity between distinct beds. It may be that the amplitude reduction effect seen in this study is largely caused by a smaller proportion of reflectors being oriented

tangential to the expanding wavefront (see Figs. 3 and 5).





We also see evidence that seismic scattering plays a role in the amplitude reduction effect. Scattering effectively increases the path length of a ray within the medium, so scattered energy will appear to arrive at a later TWTT than primary reflections. This has the effect of reducing the total energy recorded within the primary TWTT of the heterogeneous zone and increasing the amplitude at later TWTTs (a so-called 'coda'). We do observe a coda, both in the single-source and the realistic multi-source
synthetic experiments (Figs. 4 and 6). There are likely two main scattering components to this coda: i) multiple scattering (so-called 'internal multiples') and ii) diffractions from heterogeneities. The contribution of multiple scattering should be largely independent of the lateral correlation length, as it is principally energy reflected more than once quasi-vertically between beds. Diffractions are generated by heterogeneities and lateral truncations on the scale of the seismic wavelength (Schwarz, 2019). As the lateral correlation length decreases, the frequency of small heterogeneities and bed terminations (lateral truncations)
increases. This implies that the proportion of energy that is diffracted, rather than reflected, from the heterogeneous zone is increased. The consequence is an increased coda amplitude, and a decrease in the amplitude of reflections within the primary TWTT of the heterogeneous zone (as seen in Figs. 4 and 6).

Analytical techniques, such as 'convolutional' modelling, are able to predict the amplitude response of individual, isolated seismic reflectors. Our results showing the divergence of seismic amplitudes from the reflectivity indicate that more sophisti-
cated modelling techniques are necessary to properly model the seismic amplitude response of heterogeneous geobodies (as shown in, e.g., Carcione and Gei, 2016). The observed contribution of scattering to the recorded wavefield implies that full-wavefield seismic modelling techniques, as used in this study, are necessary to accurately reproduce the seismic amplitudes. In this study we develop a workflow for empirically estimating the amplitude contribution from heterogeneity (a proxy for deformation) using a random media approximation for MTDs and finite-difference full-wavefield seismic modelling.

**5.2 Seismic 'transparency' in mass-transport deposits**

We suggest that, to a first approximation, the internal heterogeneity of MTDs is similar to the random media models we use for this study. Previous studies have suggested similar models for mass-transport related stratal disruption, whereby a thin-bedded sedimentary sequence is progressively deformed by mass-transport processes, creating laterally truncated beds and ultimately a 'block-in-matrix' style fabric (Ogata et al., 2012; Ford and Camerlenghi, 2019, see Fig. 7b, this study). This
style of internal deformation is visually similar to anisotropic, two-component exponential random media (Figs. 3, 5 and 7). MTDs contain a wide variety of internal structural fabrics, and we do not aim to precisely replicate these with the random media models. Mass-transport processes in general, however, will act to laterally extend or compress (by faulting, folding and shearing) previously undeformed sediments. Stratal disruption therefore acts to increase the lateral heterogeneity, equivalent to decreasing the lateral correlation length. Even though the random media models used in this study do not accurately reproduce
realistic internal structure of MTDs, they can approximate the first-order changes in heterogeneity caused by mass-transport processes.

Furthermore, we suggest that the observed amplitude reduction effect could contribute to the 'acoustic transparency' effect often associated with MTDs in real-world seismic data. The results of these synthetic seismic modelling experiments suggest that decreasing the lateral correlation length within a heterogeneous geobody can significantly reduce the average seismic




amplitude in single-channel seismic experiments (Figs. 4 and 6). Therefore, we suggest that progressive deformation within MTDs is also likely to generate a significant average amplitude reduction in single-channel data, relative to unfailed sediments. In other words, it is likely that 'acoustic transparency' can in some cases result from stratal disruption alone, without having to invoke mechanisms to reduce the internal reflectivity such as fine-scale mixing, disaggregation or the presence of free gas. Moreover, this stratal disruption can be on a scale significantly larger than the dominant seismic wavelength (Figs. 4b and 6d).

MTDs are commonly associated with seismic diffractions in seismic profiles (Urgeles et al., 1999; Diviacco et al., 2006; Ford et al., 2021). In the random media experiments in this study, the primary source of scattering is likely to be tip diffractions from the 'bed terminations'. In real geology, simple deformation will not generate these kind of terminations—but the apexes of folds may. It is likely that real-world MTDs will have additional sources of scattering beyond those we model here with random media, including basal grooves, ramp-and-flat basal topography, remnant blocks, faulting and vertical erosive surfaces (Bull et al., 2009; Ford et al., 2021).

An alternative mechanism for reducing the observed seismic amplitudes within real-world MTDs is the removal of internal reflectivity due to disaggregation and fine-scale mixing during failure and emplacement (i.e., at scales much smaller than the seismic wavelength). Many previous studies have considered this a likely source of seismic 'transparency' (see references in Section 1). We consider that the removal of internal reflectivity is indeed commonplace within real-world MTDs. There are many documented outcrop examples of fine-scale mixing within component lithologies of the slide mass, grain-scale deformation/alteration and petrophysical changes due to densification and over-compaction (Ogata et al., 2014). Over-compaction is caused by sliding disturbing an existing pore network and removing porosity, with the effect of making the bulk mass denser (due to a lower proportion of relatively low density pore fluids), and reducing the magnitude of pre-existing impedance contrasts. This in turn reduces the effective reflectivity and therefore the seismic amplitudes. Another mechanism likely to be common in real-world MTDs is seismic attenuation, e.g., from partially saturated pore fluids (free gas). Seismic attenuation causes amplitude reduction and preferential attenuation of high frequencies. This has the effect of creating vertical 'blanking zones' below an attenuating geobody and a loss of resolution with increasing depth. Due to the relatively low seismic penetration of sub-bottom profiler data, there may be scenarios where such attenuating zones appear similar in character to seismically 'transparent' zones. MTDs are indeed often associated with fluid expulsion and subsurface fluid flow (Diviacco et al., 2006; Sun et al., 2017; Moernaut et al., 2020). Moreover, the presence of free gas in the pore space reduces the bulk density compared to water, therefore will act to reduce the internal acoustic reflectivity, on average. We consider it likely that some seismically 'transparent' zones in real-world MTDs are actually caused by free gas, rather than by other sources of amplitude reduction. In this study, however, we show that seismic attenuation is not necessary to generate such 'transparent' zones. This should be taken into account when interpreting datasets from geological settings where heterogeneous geobodies (e.g., MTDs) are found close to attenuating geobodies (e.g., gas clouds and mud volcanoes).

It should be noted that the maximum amplitude reduction observed in the realistic multi-source synthetic experiment ($\sim$15%, Fig. 6) is not as large as the amplitude reduction observed in the 2017 Black Sea geohazard survey real data example ($\sim$50%, Fig. 1). In real data the magnitude of the amplitude reduction will be strongly connected to the bandwidth of the seismic source, the signal penetration and the scale of the internal deformation. This means that even if we choose accurate elastic





and geostatistical parameters for the realistic multi-source synthetic experiment, the results are not likely to be quantitatively comparable with the real data. In addition, we model the sub-bottom profiler seismic source as a point source, with a spherical wavefront. In reality, sub-bottom profiler sources are often transducers with dimensions that are a non-negligible fraction of the dominant source wavelength. This can be used to create a 'beam-forming' effect, which reduces the signal-to-noise ratio and increases the penetration by concentrating energy in a narrower, focused beam. We suspect that the real-world amplitude

reduction effect may in fact be larger with such a source geometry due to the smaller effective Fresnel radius.

The single-source synthetic experiment results suggest that it is likely that this amplitude reduction effect is not strongly controlled by the relationship between the dominant wavelength and the average bed thickness (Section 5.1). This means that the deformation-induced amplitude reduction effect likely generalises to many geological settings, in different water depths and with different seismic source bandwidths. These results imply that simple 1-D models for wave propagation in MTD-

style geology are not appropriate even for single-channel, zero-offset data. Several previously published works have made the implicit assumption that seismically 'transparent' MTDs are associated with little preserved internal structure (Section 1). The results of this study show that an amplitude reduction effect can be generated even by MTDs that preserve coherent internal structure around the scale of the seismic wavelength, and significantly above core-scale (Figs. 6 and 7). We argue, therefore, that 'acoustic transparency' does not necessarily correspond to a lack of well-preserved internal structure around the scale of

the seismic wavelength. Nevertheless, it is likely that in the real-world this effect is a combination of deformation-induced amplitude reduction (this study), true reflectivity loss from mass-transport processes and possibly an attenuation component where the MTD is associated with free gas.

Finally, we also observe a small amplitude *increase* effect in the analysis window below the MTD (Fig. 6). We speculate that this increase could be due to contributions from i) lower transmission losses in the MTD zone with increasing deformation, in-

creasing the amount of energy reflected from beneath; ii) scattering from heterogeneities not directly below the source/receiver (i.e., delayed arrivals of point-scattered 'diffraction tails'); or iii) multiple-scattered arrivals within the heterogeneous layer (internal multiples recorded at later TWTT compared to the primary reflections). Lower transmission losses are not relevant in the single-source experiment, because there is no reflectivity within the homogenous layer beneath the heterogeneous zone. Contributions from off-vertical scattering are relatively small, as the recorded trace in the unfailed sediments does not appre-

ciably change between the pre- and post-failure models, indicating that the MTD zone is not generating scattering to strongly affects the seismic image outside it (Fig. 6a-c). It is plausible that when the geological properties are very well constrained, the amplitudes *below* an MTD could be used in the future to differentiate between amplitude reduction due to internal deformation and a true lack of internal reflectivity. The synthetic modelling performed in this study should be supplemented in future by more sophisticated and realistic seismic source modelling.

## 5.3 Core-scale characterisation of mass-transport deposits

Fig. 7a reproduces virtual cores extracted from the MTD zone in the realistic multi-source synthetic experiment, with progressively increasing degrees of deformation. At core-scale, deformation structures are only noticeable when lateral correlation lengths are below approximately $a_x \leq 0.5$ m (Fig. 7a). The seismic amplitude reduction effect, however, begins to be notice-





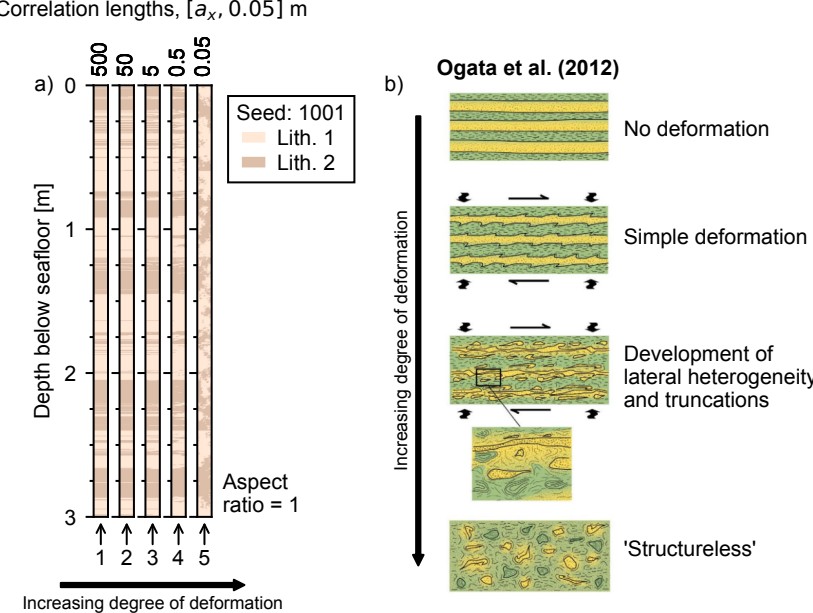

**Figure 7.** a) Virtual cores from the mass-transport deposit zone in the realistic multi-source synthetic experiment (Fig. 5), with progressively decreasing lateral correlation lengths $a_x = \{500, 50, 5, 0.5, 0.05\}$ m and vertical correlation length $a_z = 0.05$ m. b) Conceptual cartoon model for progressive stratal disruption in a thin layered two lithology sedimentary sequence, from unfailed, undeformed strata to a strongly deformed 'block-in-matrix' style fabric (modified from Ogata et al., 2012).

able at much larger lateral correlation lengths, below approximately $a_x \leq 10$ m (Fig. 6b). In other words, for progressively
increasing deformation, the amplitude reduction effect can be observed in the seismic data well before deformation is visible in
sediment cores. This result is not surprising—the effective maximum 'lateral resolution' of cores (up to $\sim 0.1$ m) is an order of
magnitude lower than the effective minimum lateral resolution of typical single-channel seismic surveys (metre-scale, Vardy,
2015).

Core-seismic correlation is possible in many sedimentary environments that have parallel and undeformed bedding (i.e.,
could be represented by random media with very long lateral correlation lengths). Core-seismic correlation implicitly relies on
a simple 1-D model of wave propagation in sedimentary sequences. When bedding is strongly deformed, lateral heterogeneity
increases and the lateral seismic resolution becomes significant. These modelling experiments show that seismic amplitudes
can be significantly affected by lateral heterogeneity. This means that core-seismic correlation is likely to be unreliable in
geobodies containing strongly deformed sediments. Therefore in general it should not be surprising that very often core-
seismic correlation within MTDs is challenging (e.g., Expedition 316 Scientists, 2009; Strasser et al., 2011; Sammartini et al.,
2021, Fig. 2, this study).



## 6 Conclusions

In this study we investigate the seismic amplitude response of two-component, anisotropic random media to changing heterogeneity using 2-D elastic finite-difference modelling. We suggest that this type of random media model may be a reasonable approximation of the heterogeneous internal structure of MTDs, where previously distinct strata have been strongly deformed and disrupted by mass-transport processes.

The single-source synthetic experiment shows a sustained decrease in seismic amplitudes with decreasing lateral correlation length, replicated across a two-orders-of-magnitude range of vertical correlation lengths. The realistic multi-source synthetic experiment, designed to replicate an AUV-style sub-bottom profiler acquisition over a realistic MTD scenario, also shows a sustained decrease in seismic amplitude with decreasing lateral correlation length. The elastic properties of the two component lithologies are fixed, thus the magnitude of impedance contrasts remains constant for all realisations. These results indicate that a significant reduction in seismic amplitude within MTDs can be caused by straightforward deformation, rather than either petrophysical changes from mass-transport processes (e.g., fine-scale mixing, overcompaction or pore fluid substitution) or seismic attenuation. The magnitude of this amplitude reduction depends on many acquisition and geological parameters, therefore the results are not quantitatively comparable with real-world data. We do suggest, however, that this numerical modelling evidence should dissuade practitioners from making strong claims about the internal structure of MTDs based on their seismic amplitude response alone. Put simply, 'acoustic transparency' does not necessarily imply a lack of coherent internal structure around the scale of the seismic wavelength, because the seismic response of MTDs is strongly controlled by the geometry—in addition to the magnitude—of the internal reflectivity. Additionally, this reduction in seismic amplitude can, according to these modelling experiments, be generated by lateral heterogeneity that is much larger than the diameter of sediment cores (10s of centimetres). Apparently undeformed sediment cores are not incompatible with seismically 'transparent' MTDs.

*Code and data availability.* Python code to generate the models, run the seismic forward modelling and reproduce Figs. 3 to 7 is available at Ford et al. (2022). Model building uses Numpy (Harris et al., 2020) and Scipy (Virtanen et al., 2020), and GSTools is used to generate the binarised random fields (Müller et al., 2022). The seismic forward modelling uses Devito (Louboutin et al., 2019), and results are visualised using Matplotlib (Hunter, 2007).

*Author contributions.* JF: Conceptualization, Investigation, Methodology, Software, Visualization, Writing – original draft; AC: Funding acquisition, Supervision, Writing – review & editing; FZ and MC: Data curation, Resources, Supervision, Writing – review & editing (CRediT – Contributor Roles Taxonomy).

*Competing interests.* The contact author has declared that neither they nor their co-authors have any competing interests.



*Acknowledgements.* Field data presented in Figs. 1 and 2 and in the Supporting Information are used with permission from RINA Consulting. J. Ford was supported by a Marie Skłodowska-Curie Doctoral Fellowship through the SLATE Innovative Training Network within the European Union Framework Programme for Research and Innovation Horizon 2020 under Grant Agreement No. 721403.



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
