# Peer review of "Seismic amplitude response to internal heterogeneity of mass-transport deposits"

_EGUsphere, 2022_

## Referee Comment (RC2)

[referee-annotated manuscript omitted]

---

## Author Comment (AC2)

Dear Dr Pilia,

Thank you for handling the editing of this submission. We have read the comments from the two reviewers and revised the manuscript accordingly. I have uploaded an updated version of the manuscript and a 'track changes' version alongside this response letter. Please see below our detailed responses (in red) to the reviewers' comments, accompanied by a list of changes from the original submission.

In the revised submission, we have taken advantage of additional high-performance computing resources to re-run the original experiments at higher resolution and adjusted the parameters of both the single- and multi-source synthetic experiments to be more relevant to the Black Sea case study example. I draw your attention, in particular, to the revised Supporting Information, where we have included new data and several new modelling experiments which we hope will address some of the reviewers' concerns surrounding how representative the models are, and the generalisability of the reported amplitude reduction effect.

I would like to take the opportunity to thank both reviewers for taking the time to read the submission and for providing constructive feedback.

Kind regards,
Jonathan Ford (on behalf of the authors)

Note: line numbers refer to the 'track changes' version

**Reviewer #1 (Jasper Moernaut)**

This is an overall excellent study that deals with an important topic in seismic stratigraphic analysis: i.e. the identification and interpretation of mass-transport deposits. The data leads to an important suggestion: "this numerical modelling evidence should dissuade practitioners from making strong claims about the internal structure of MTDs based on their seismic amplitude response alone." I agree with the authors that many studies make unsupported interpretations based on the internal facies of MTDs on single-channel seismic profiles. This manuscript forms a good scientific base that helps stopping this practice. Moreover, it explains the apparent discrepancy between the MTD seismic facies (transparent/chaotic) on seismics and the rather undeformed/intact sediment sequences observed in cores taken in such MTDs. This is indeed a very common observation that puzzled many sedimentologists and hampered the identification of MTD upper and lower limits on sediment cores. Overall, it is a very important research question and the authors present an original approach to address it.

The manuscript is very well written, adequately structured and contains clear illustrations. The methodological approach (synthetic 2-D elastic seismic modelling) is well described and its implications for a broader marine geoscientific audience are clearly presented. The fact that the study contains both simulations as well as data from a natural case study makes it suitable for a broad audience. The authors extensively discuss which other factors may contribute to the low-amplitude seismic facies of MTDs and adequately place their results in this open discussion.

I have a few minor remarks and discussion points for which I would like to see the opinion of the authors. Possibly a few extra explanations in the manuscript could make such technical points more accessible for a broader audience. Overall, this is an excellent manuscript that can be accepted for publication after some minor revisions.

Thank you for the positive comments and constructive feedback.

**Comments**

Sediment properties: The authors write that their simulation results are not quantitatively comparable with real-world data. However, the simulated mass-transport deposit uses "realistic" sediment properties obtained by sediment core analysis in the lab. In my opinion, the P-wave velocity measurements in the lab ( < 1400 m/s; Figure S1) are probably not representative for in-situ conditions. The problem with coring of gas-rich sediments is that gas expends when retrieving the cores. This leads to crack formation and thus much reduced P-Wave velocities measured in the lab. This implies that the values derived by lab measurements on cores can be very different than values of the MTD in nature (gas is rather dissolved or present as small bubbles; no cracks in the natural sequence). Such cracks could explain the noisy P-wave velocity data. However, the density data does not show cracks and high values of about 2 kg/m3. This leads to a strange combination of much higher density and much lower P-wave velocity compared to water.

We agree that the seismic velocities measured on the sediment cores are not properly representative of in situ conditions. We originally used the core measurements to reproduce geologically plausible contrasts in the physical properties of the multi-source model (as the P-impedance contrasts, rather than the absolute P-impedance values, are the primary control on the seismic amplitudes). In the revised manuscript, we have maintained the contrast in seismic velocities based on the MSCL measurements, but shifted the minimum P-wave velocity to match seismic velocities from a nearby surface multi-channel seismic profile (Fig. S3). Our hope is that this provides slightly more representative in situ physical properties, even if the primary control on the amplitude of reflections is the average P-impedance contrast, which is unchanged. We agree that it is incorrect to call these "realistic" sediment properties, and this may be confusing to the reader. Now that the values are more plausible (i.e., not immediately confusing to the reader), we have unified the sediment properties between the single- and multi-source experiments for consistency, and removed references to "realistic" sediment properties where approriate.

- Renamed the "realistic multi-source experiment" to "multi-source experiment" throughout the text

- Added a nearby stacking velocity profile (with Dix-converted interval velocities) to the Supporting Information (Fig. 1a, Fig. S3)

- Updated the water velocity and the sediment velocities (Table 1)

  - Re-ran the single-source experiment (Fig. 4)

  - Re-ran the multi-source experiment (Fig. 6)

- Revised the text to emphasise that we are interested in reproducing (quasi-)realistic P-impedance contrasts (as these are the primary control on the amplitude response), not necessarily in replicating precise elastic parameters (L93, L121)

A similar comment can be given for the S-wave velocity data used (Table 1): Maybe justify a bit more why you assume these are 0.5x the P-wave velocity? Some comparison with measured S-wave velocity on shallow sediments in other study areas would be useful.

As we don't have any information on the S-wave velocity of these sediments, the choice of v_p/v_s=2 was arbitrary, based on typical values used for consolidated sediments in hydrocarbon exploration. These values are likely unrealistic for poorly consolidated marine sediments like these, indeed (see our response to Reviewer #2). On the other hand – MTDs are often significantly over-densified compared to the source sediments, so we feel that predicting the S-wave velocities based on examples of non-MTD shallow marine sediments may be equally unrealistic. We are not aware of any direct, in situ seismic-scale measurements of v_s on unburied, shallow MTD sediments similar to our Black Sea case study.

The v_p:v_s ratio is not critical here because we are modelling single-channel/zero-offset data, where the contribution of energy partitioning from non-normal incidence should be (very) small. Therefore the dominant control on the seismic amplitudes will be the P-impedance contrasts. To check this, we've performed a separate single-source synthetic experiment, included in the Supporting Information (Fig. S14). The results show that a v_p:v_s=4 (Poisson's ratio 0.47) makes a negligible difference to the recorded amplitudes compared to the original v_p:v_s=2 (Poisson's ratio 0.33) (Fig. S15). We therefore expect that for the multi-source experiment any amplitude drop effect would be independent of the Poisson's ratio (i.e., independent of the degree of consolidation of the sediments).

- Changed the v_p:v_s=4 for both the single- and multi-source experiments (Table 1)

- Added a v_p:v_s=2 single-source experiment in the Supporting Information (Table S4 and Fig. S14)

  - Added a comparison plot between recorded amplitudes at v_p:v_s=4 and v_p:v_s=2 to the Supporting Information (Fig. S15)

- Added a discussion of the choice of the v_p:v_s with reference to typical Poisson's ratios found in shallow marine sediments (L107-L120)

So, even though it is not the purpose of the authors to fully recreate the natural conditions from the study area in the simulations (line 110-114), I think it could be valuable to also run the simulations using other (more realistic?) sediment property values obtained from other studies, and compare the outcomes.

Unfortunately the computational cost of modelling the multi-source synthetic experiment is very high, so it is not feasible to run many runs to test the impact of different sediment properties. This is something, however, that can be tested easily on the single-source synthetic example, as the models are much smaller and the runtimes much more feasible. We include results of the single-source synthetic experiment with significantly lower (Fig. S10) and significantly higher average (Fig. S11) impedance contrasts in the Supporting Information, for comparison. They both show a similar

magnitude (proportional) reduction with decreasing lateral scale length effect to the original single-source synthetic experiment (Fig. S15).

- Added the approximate computational times for each experiment for reference (Table S5)

- Added 'low reflectivity' (Table S1 and Fig. S10) and 'high reflectivity' (Table S2 and Fig. S11) single-source synthetic experiments to the Supporting Information

Figure 1: It is very illustrative to see these profiles and the quantification of RMS envelope amplitude.

Figure 6d: Reduction in RMS amplitude for smaller lateral correlation lengths: your results show a clear decrease in RMS amplitude, but this is not strong enough to explain the rather transparent MTD facies found in natural case studies. Indeed, the Black see study shows 15% amplitude reduction in simulations and 50% on the seismic profile. (L293). The authors did a great job in explaining the possible reasons for the different value of this amplitude reduction effect.

Note that in the revised manuscript the multi-source experiment has been re-run with a dominant source frequency of 1.5 kHz – the amplitude reduction is now on the order of ~25% (text updated).

L131: Why do you use a 1.1 kHz frequency for the recreated AUV acquisition, whereas you use 3.5kHz for the single-source synthetic experiment? Why not using the same value for both simulation experiments? Would it give a big difference in the results?

3.5 kHz was simply chosen as a common SBP dominant frequency for, e.g., pinger sources, as it will be more familiar to practitioners. For the original submission we had to limit the multi-source experiment dominant frequency to 1.1 kHz, as this was the maximum frequency that it was computationally feasible to model in a model this large. The primary control on the compute time is actually the minimum S-wave velocity: the maximum grid spacing to avoid numerical dispersion depends on the minimum velocity in the model. Since the submission, we have gained access to more HPC resources, and we have been able to re-run the multi-source experiment at higher resolution/higher source frequency of 1.5 kHz (unfortunately 3.5 kHz is still unfeasible) with a $v_p : v_s = 4$. We agree that having all the experiments with the same source frequency is preferable, so we have adjusted the single-source experiments to also have 1.5 kHz dominant frequency. The revised results are comparable to the original experiments.

- Re-ran the multi-source and single-source experiments with a dominant source frequency of 1.5 kHz (Table 2, Figs. 4 and 6)

- Included estimates of the compute time in the Supporting Information (Table S5), for reference

L 157: Coda below the MTD: this conclusion would get stronger if you can support the existence of a coda below MTDs with observations on seismic profiles of natural case studies. In my experience (MTDs in lakes measured with a 3.5kHz pinger), there is no clear coda visible below (e.g. see the supplementary data in Praet et al., 2017. https://doi.org/10.1016/j.margeo.2016.05.004). In gas-rich settings, high amplitude reflections and acoustic turbidity is common below (or within) the MTDs (e.g. Moernaut et al., 2020. GeolSoc) and thus difficult to assess whether a coda is present or not. Can you find some examples in literature (gas-free settings) that exhibit this coda below MTDs?

The coda is a tricky one. The explanation is fairly straightforward – if the MTD is reflecting less primary energy this energy has to go somewhere. Either it can be absorbed, or it can be multiple energy, or it can be off-vertical reflections/diffractions, or more energy is transmitted through the MTD to generate brighter reflections below. All of these imply apparently higher amplitudes at a later TWTT. Whether we can see this coda in real world data – we suspect unlikely, as (like you say) MTDs are so strongly associated with complex topography, noise, attenuation effects and high amplitude reflectors below the MTD (e.g., in Praet et al., 2017) which in practice would obscure it. Plus the standard SBP data processing flow includes a bunch of non-amplitude preserving filters etc. We thought to report this as a curiosity coming from the synthetic modelling.

- Added a paragraph acknowledging the problems of detecting a coda in real world data to the Discussion (L393-L398)

L165: Realistic multi-source synthetic experiment: these calculations were made for a seismic source at 40 m above the bottom. Many studies on MTDs are with hull-mounted systems or towed systems (on lakes). So how would an larger water column between seismic source/receiver and the sea bottom affect your conclusions? The Fresnel zone will get larger. Will this have an effect on your outcomes?

Good question, and one that is difficult to answer with synthetic modelling for the multi-source case, for computational reasons (see above). As you say, in reality the source is very often hull-mounted or towed so it's much further from the target. The only reason why it is computationally feasible to model this Black Sea case study at quasi-realistic source bandwidths is because the AUV source is so close to the target MTD. We can, however, easily test the effect of increasing the source-target distance (i.e., a larger Fresnel zone) for the single-source case. We have presented a 'far source' example for the single-source case in the Supplementary Information (Fig. S12). We see lower recorded amplitudes (due to geometric spreading) but the 'amplitude drop' effect has a similar magnitude as for the original single-source experiment (~25%).

- Added a 'far source' single-source synthetic experiment (Table S3, Figs. S12 and S13)

- Added a discussion of the likely impact of a more distant source to the text (L283-L298, L365-L369)

L213-214: "This also means that the heterogeneity-induced amplitude reduction effect should not be strongly dependent on the dominant wavelength of the seismic source." OK, but the wavelength of the seismic source has an influence on the lateral resolution of the data (Fresnel zone), so I guess this should also affect the amplitude reduction. Please explain why this would not be the case. Maybe I misunderstood something here.

We meant to say that the existence of this heterogeneity-induced amplitude reduction effect should be seen across a wide range of seismic bandwidths, for a given geological scenario. The magnitude of the effect will likely be coupled to the dominant wavelength, exactly for the reasons you describe. We now clarify this in the text (L283-L298), and mention the new single-source examples that we present in the Supplementary Information.

L220: Yes, I also do think that the distribution of the dips of the interfaces within the heterogeneous zone can be a major reason for the amplitude reduction effect. Moreover, many natural examples of MTDs show faulting of rather coherent (horizontally-stratified) sediment units due to extension (source area) or compression (toe area). For example, a strong horizontal reflector (sand layer) can be positioned next to homogenous mud. So for a given two-way travel time, the Fresnel zone can contain a mixture of signal interference (positive, negative, in between) and the total reflection amplitude will be lower than for a continuous horizontal reflector. Maybe such offset between MTD blocks should be mentioned in the article. A natural example can be found in Sammartini et al., 2021 (https://doi.org/10.1016/j.sedgeo.2021.105877). This is especially relevant for frontally confined landslides and blocky zones due to headscarp retrogression.

Good point. We have added a sentence acknowledging the particular importance of the Fresnel zone in the MTD case (L364-L365).

Good luck with addressing these few comments!

**Reviewer #2 (Martino Foschi)**

The author investigated on the acoustic reflectivity of a mass transport deposit (MTD) from an area offshore the Blake Sea. The database is composed of geophysical data and core data. The latter intersects the shallower section of the 35 metres' thick MTD. The core data shows clear stratification, while the geophysical data exhibits acoustic transparent facies. Acoustic transparency is known in literature, and it is generally associated with lack of structures or stratification. The authors build a number of geological/elastic models that are consistent with a general view that MTDs are

characterized by an internal featureless (random) character. The models are then recorded using single and multi-source acquisition synthetic experiments (forward modelling) and the resulting images show that the more random character MTDs are characterized by lower amplitude and no coherent reflectivity.

Thanks for taking the time to read the manuscript and for providing feedback.

We would like to point out that the term "random media" (a mathematical description) is not the same as the concept of geological randomness (heterogeneity). Whilst 2-D binarised random media can be strongly (or weakly) heterogeneous, they can also be homogeneous in the extreme case of very long scale lengths, or more commonly homogeneous in one dimension (e.g., very long lateral scale lengths – see some of the "pre-failure" examples in the study). We simply use random media to represent a range of geological scenarios between stratified, well-bedded geology (very long lateral scale lengths), and very heterogeneous geology (the "isotropic" case where the lateral and vertical scale lengths are equal).

While we do observe that models with a lower lateral scale length generate a lower average seismic amplitude response, we do not assess the degree to which there are still coherent reflections. Our main claim is that an appreciably lower amplitude response can be generated even when there is still coherent (metre-scale) subsurface reflectivity preserved.

On my view the authors and the method used fail to really address the actual significance of the acoustic character of MTDs and study produce little advancement on the understanding of MTDs in general. The elastic models are what the authors believe the internal characteristic of an MTD is. The results are compatible only on those proportion of MTDs where the model assumptions are met. MTDs are far more complex features, characterized often by region of compression and extension, with faults, folding, residual stratigraphy of the transported blocks and so on (the authors seem to be aware of that) and these features are often observed on geophysical data.

We make specific and limited claims about the effect of lateral heterogeneity on the average amplitude of the seismic amplitude response only, not on the general seismic/acoustic character (e.g., seismic facies) or on any structures which may or maybe not be interpretable in real world data. In the real world, faults and folding etc all contribute to reducing the lateral scale length of the geological heterogeneity within an MTD compared to unfailed sediments. The fact that some of these structures are at a scale large enough to be imaged by seismic experiments doesn't preclude that there are equivalent structures below the effective seismic resolution which still have a strong control on the amplitude response.

The idea of the modelling is to create an extremely simplified model of heterogeneous geology (perhaps similar to some MTDs) in order to test the specific contribution of increasing lateral heterogeneity to the average amplitude response. We don't aim to replicate realistic internal structure of MTDs. The argument then follows that complex internal structure IS lateral heterogeneity, so we would expect to see similar amplitude effects in real MTDs with real internal structures.

- Removed references to the "realistic" multi-source experiment throughout the text to avoid giving the impression that we are trying to recreate real internal structure of MTDs

Instead of proposing a generalized model for the internal characteristics of MTDs in general the authors should investigate on the character of the MTD in this study. The MTD is stratified, so the authors should better address the fact that a stratified media can still be nearly acoustic transparent. Also, is the binary model proposed not compatible with the studied MTD? Does the depositional environment mixed most of the lithologies so that all the thin layers observed in the core are characterized by similar elastic parameters? The model, in my view, should benefit more from core data, which intersect and sample the MTD for at least 15% of the actual thickness. The core information is expected to extend a few meters horizontally over the actual cored surroundings and provide a small, but real, depositional character of what the MTD is.

We do not propose a generalised model for the internal characteristics of MTDs. The MTD in the study is merely included as a real world example of an MTD that appears to be stratified at core-scale, whilst being "seismically transparent" at sub-bottom profiler scale. We do not claim that the core observations should or can be extrapolated to the whole MTD.

The binarised random media model was not chosen to be compatible with the case study MTD, it is merely used to build a model for heterogeneity that i) is parameterised by a small number of parameters ($a\_x$ and $a\_z$) and ii) maintains approximately constant average reflectivity (by using only two distinct lithologies instead of continuously varying parameters).

The results of the seismic modelling experiments show nicely why you can't extrapolate core properties laterally in strongly heterogeneous/deformed geology. See Fig. 7a, with an example of a synthetic core with lateral scale length $a\_x=0.5$ m (i.e., on average this view can only be laterally extrapolated on the order of 0.5 m) which still appears fairly well stratified at core-scale.

Otherwise, the manuscript reads quite well, and it is very easy to follow. The figures and the data used are of great quality. The method section could be shorter with most of the model description transferred into a table. Conversely, the tables contain very little information and could be written along the text. The discussion should be rewritten based on the results. More detailed points have been added to the attached pdf.

Agreed, we have made the Methodology section more concise by better using the tables.

- Merged (previous) Tables 1 and 2 – the single- and multi-source experiments now use the same elastic parameters for the component lithologies (see response to Reviewer #1)

- Moved most of the modelling parameters for the single- and multi-source experiments from the text to a (new) Table 2

**Detailed comments (annotations from attached PDF)**

L26 Most MTDs observed on seismic data preserve plenty of reflectivity, with kinematics indicators, reflectivity from intact blocks, faults, folding structures, etc. The author seems to know this as stated in the next sentences…

We agree – at least for many MTDs imaged by modern, 3-D, multi-channel data. In this study, however, we deal with the case where seismically imaged MTDs do not show strong or coherent internal seismic reflections (so-called "acoustically transparent" MTDs). This situation is still common, especially in MTDs imaged by sub-bottom profilers, vintage seismic data, 2-D data etc. Note that simply preserving internal reflectivity (i.e., internal structure, kinematic indicators etc) does not necessarily mean that the MTD will show coherent seismic reflections (the main claim of this study).

- Added "in both single- and multi-channel data" (L28)

L37 the variation of acoustic impedance with depth, which dictate the polarity of the MTD, or a portion of it, reveals whether the MTD is more or less compacted than the sounding formation. It is straightforward (e.g., Mavko et al., 2009, Rock physics Handbook)

It is true that the acoustic impedance variation with depth straightforwardly controls the amplitude and polarity of seismic reflections – but only in 1-D/layer cake geology and with normal-incidence reflections. This may be a reasonable approximation for far-field reflections from parallel and horizontal unfailed sediments, but when the internal structure of MTDs is strongly deformed it is not safe to assume this any more (as our modelling results show later).

Note that most SBP data is displayed in envelope trace (as is the data in this study) – so polarity is not relevant. In this study we deal entirely with the absolute amplitude, and do not consider any polarity effects.

Figure 2. is the stratification here associated with multiple stacked MTD? or is the primary depositional character?

We suggest that this is a single MTD, based on the external morphology, the (self-)consistency of the internal seismic response and the lack of clear erosional surfaces within the MTD interval in the core. It is, however, completely possible that the MTD (or rather, MTC) is a composite of several events, which could indeed be one cause of the stratification. The source of the internal stratification isn't important to this study, just that within the acoustically transparent MTD zone there is little obvious evidence of deformation structures.

L90 what is the phase of the pulse (e.g., minimum, zero, etc)

It's a zero-phase Ricker wavelet

- Added the polarity to Table 2

Table 1. is the sediment consolidated? what do you expect the elastic moduli to be in this setting (recently deposited MTD)?

We do not have good data to judge how well consolidated the MTD and the surrounding sediments are, so we assumed a constant v_p:v_s ratio everywhere. That said, the previous v_p:v_s=2 implies that these are sediments are very consolidated/lithified, and very much too low. We've since used v_p:v_s=4, and added a short discussion in the manuscript about this choice (which is still arbitrary), and added a single-source experiment to the Supporting Information to test the impact of changing the S-wave velocities (Fig. S14). We see a negligible difference in the modelled amplitudes between the "high" and "low" Poisson's ratio scenarios, because for marine zero-offset data the P-impedance is the dominant control on the amplitude (very little mode-converted energy is reflected back) (Fig. S15). Please see also our response to similar comments from Reviewer #1 regarding the elastic moduli.

- Changed the single-source experiment to use the same elastic parameters as the multi-source experiment (Table 1)

- Changed the v_p:v_s=4 for both the single- and multi-source experiments (Table 1)

- Added a v_p:v_s=2 single-source experiment in the Supporting Information (Table S4 and Fig. S14)

   ○ Added a comparison plot between recorded amplitudes at v_p:v_s=4 and v_p:v_s=2 to the Supporting Information (Fig. S15)

- Added a discussion of the choice of the v_p:v_s with reference to typical Poisson's ratios found in shallow marine sediments (L107-L120)

L121 this is not what the author show in the core data. Why not replicate the model so that the system is stratified but where the individual beds are characterized by similar elastic character.

The binarised random medium (i.e., dividing the geology into two lithological end-members) is merely to maintain approximately constant impedance contrasts in the medium, we are not suggesting that this is to replicate the distribution of P-wave velocities and densities seen in the core. We have added a version of the single-source experiment with much lower (x0.25) and much higher (x4) P-impedance contrasts, and we still see a similar magnitude amplitude reduction effect (Fig. S15).

- Added a 'low reflectivity' single-source experiment to the Supporting Information, where the component lithologies have more similar elastic properties (i.e., reflectivity is low) (Fig. S10)

- Added a 'high reflectivity' single-source experiment to the Supporting Information, where the component lithologies have much less similar elastic properties (i.e., reflectivity is high) (Fig. S11)

Table 2. could you please discuss on the expected porosities of these sediments and how this parameter can affect the bulk modulus first and the compressional and shear velocities?

We don't have any (direct) information about the porosity of these sediments. As you mention above, within an MTD the situation can be complex due to over-compaction. As pointed out by Reviewer #1, the measured P-wave velocities here are not representative of in situ conditions, as the porosity is presumably high enough (possibly also through cracks induced by gas expulsion) that the MSCL P-wave velocities are too low. We've added a stacking velocity profile from the area and adjusted the P-wave velocities here based on this to be more representative of the in situ conditions. See our response to Reviewer #1 for more details.

- Added a nearby stacking velocity profile (with Dix-converted interval velocities) to the Supporting Information (Fig. 1a, Fig. S3)

- Updated the water velocity and the sediment velocities (Table 1)

  - Re-ran the single-source experiment (Fig. 4)

  - Re-ran the multi-source experiment (Fig. 6)

Figure 3. this model is far from a geological concept of MTDs where there are region of compression and extension, region composed of intact blocks floating within a complex matrix made of moving sediments and etc. I am wondering where does this model apply within an actual MTD?

We do not claim that this is a realistic model for the internal structure of a real world MTD. It is true, however, that compression, extension, blocks etc all act to increase the lateral heterogeneity within a MTD (i.e., decrease the "lateral scale length"), relative to unfailed sediments. The aim of this study is to test the effect of reducing the lateral scale length of heterogeneity on the seismic amplitude response.

Nonetheless – one can imagine, for example, an MTD where i) the headwall zone has very strong lateral heterogeneity (low lateral scale length), due to an abundance of extensional faulting, ii) the translational zone has a relatively weaker lateral heterogeneity, as it is composed more of translated but largely intact blocks, and iii) the compressional zone at the toe of the MTD again has strong lateral heterogeneity due to lots of reverse faulting. The idea of the random media is that they can represent a wide range of possible lateral scale lengths.

Figure 5. is this model realistic of what an MTD actually is?

We do not claim that this is a realistic model for the internal structure of a real world MTD (see above).

L254 Great (see my previous points)! So, what is the scale of these changes, geologically speaking, with respect to the dominant frequency and sampling interval of this model?

Real-world internal structure (i.e. the heterogeneity) of MTDs has been reported from cm-to-km scale (e.g., Ogata et al., 2016), it's hard to say what the characteristic scale of MTD internal structure should be as it depends on the dimensions of the MTD and on the degree of internal deformation.

The sampling interval of the model is deliberately much smaller both than the dominant source wavelength (to avoid numerical modelling artefacts and to properly represent dipping interface with respect to the dominant wavelength) and than the scale of the heterogeneity (this can be verified by checking Fig. 7). In Figs. 4b and 6d we display the dominant source wavelength for comparison with the plotted average amplitudes. It's clear that there is an appreciable amplitude drop before the lateral scale length is reduced to the order of the seismic wavelength (and also after).

L340 I agree that the cores sample a small proportion of an MTD, however there are regions of MTDs where the transport is achieved along a decollement surface with minimum deformation.

We agree that in some (maybe even many) MTD scenarios there are large parts of the body that have been transported with little internal deformation (often referred to as 'megablocks' in the literature). These megablocks will have a similar internal seismic character to unfailed sediments, almost by definition. We only deal with cores within MTDs/parts of MTDs that show a low-amplitude seismic character – i.e. where there has been some kind of appreciable alteration to the sediments from mass-transport processes. In a scenario where a core has been take from within an undeformed or weakly deformed megablock, we would expect that the seismic response is similar to the unfailed sediments.